# Knockdown of *p*-Coumaroyl Shikimate/Quinate 3′-Hydroxylase Delays the Occurrence of Post-Harvest Physiological Deterioration in Cassava Storage Roots

**DOI:** 10.3390/ijms23169231

**Published:** 2022-08-17

**Authors:** Qiuxiang Ma, Jia Xu, Yancai Feng, Xiaoyun Wu, Xinlu Lu, Peng Zhang

**Affiliations:** 1National Key Laboratory of Plant Molecular Genetics, Center for Excellence in Molecular Plant Sciences, Chinese Academy of Sciences, Shanghai 200032, China; 2University of Chinese Academy of Sciences, Beijing 100049, China

**Keywords:** cassava, PPD, C3′H, scopoletin, scopolin

## Abstract

Cassava storage roots are an important source of food, feed, and material for starch-based industries in many countries. After harvest, rapid post-harvest physiological deterioration (PPD) reduces their palatability and marketability. During the PPD process, vascular streaking occurs through over-accumulation of coumarins, the biosynthesis of which involves the key enzyme *p*-coumaroyl shikimate/quinate 3′-hydroxylase (C3′H). Repression of *MeC3′H* expression by RNA interference in transgenic cassava plants caused a significant delay in PPD by decreasing scopoletin and scopolin accumulation in field-harvested storage roots. This study demonstrates that MeC3′H is the key enzyme participating in coumarin biosynthesis during PPD and shows that *MeC3′H* is a useful target gene for editing to prolong the shelf life of cassava storage roots.

## 1. Introduction

Cassava (*Manihot esculenta* Crantz) is a staple food and source of income for almost 800 million people in tropical and subtropical countries [1]. Its storage roots (SRs) have high starch content but undergo rapid post-harvest physiological deterioration (PPD) within 24–72 h after harvest, which leads to a reduction of palatability and marketability [2]. PPD involves a quick oxidative burst occurring within 15 min at damaged sites of cassava SRs [3], followed by altered expression of genes involved in antioxidant defense, together with the increased activity of a range of enzymes including phenylalanine ammonia-lyases and peroxidase and the accumulation of phenolic secondary metabolites [4,5]. The vascular discoloration occurring during PPD results from the oxidation of complex polyphenolics (such as the hydroxycoumarins scopoletin and esculetin) and their glucosides (i.e., scopolin and esculin) [6,7,8,9,10,11], which are synthesized through the hydroxycoumarin biosynthetic pathway [12]. The content of scopoletin, which is by far the most abundant hydroxycoumarin, increases by 150–200-fold within 24 h [8,13,14,15].

Three hypothetical pathways for the biosynthesis of scopoletin and other hydroxycoumarins have been proposed in various plants, namely the 2′,4′-dihydroxycinnamate, 2′,4′,5′-trihydroxycinnamate (6′-hydroxycaffeate), and 4′-hydroxy-3′-methoxycinnamate (ferulate) pathways [16,17]. The ferulate pathway has been verified to be the major pathway for scopoletin and scopolin synthesis in cassava SRs during PPD; specifically, the *p*-coumaric, caffeic, and ferulic acids are *o*-hydroxylated, isomerized, and lactonized into scopoletin and then glucosylated into scopolin [17]. The membrane-associated cytochrome P450 family protein *p*-coumaroyl shikimate/quinate 3′-hydroxylase (C3′H) (CYP98) is the key enzyme in the ferulate pathway and functions by 3-hydroxylating *p*-coumaroyl derivatives into the corresponding caffeic acid conjugates [18,19]. C3′H (CYP98) has a higher turnover rate (600/min with shikimate as a substrate) compared with other P450 oxygenases and has different substrate preferences or specificities in different species; most C3′H enzymes metabolize shikimate esters of *p*-coumaric acid more efficiently than quinate esters [19,20,21,22]. The preferred substrate of CYP98 is usually *p*-coumaroyl-shikimate even though some C3′H enzymes can hydroxylate other *p*-coumaroyl derivatives in red clover and *Arabidopsis thaliana* [22]. A cloned sweet basil C3′H (CYP98A13) is found to efficiently hydroxylate both *p*-coumaroyl-shikimate and quinate but hydroxylated 4-hydroxyphenyllactate ester with less efficiency [20]. The C3′H from *Lithospermum erythrorhizon* has the ability to hydroxylate *p*-coumaroyl-4-hydroxyphenyllactic acid, but its activities on shikimate and quinate *p*-coumaroyl ester have not been determined [23]. There are two C3′H enzymes in coffee, CYP98A35 and CYP98A36. CYP98A35 can act on *p*-coumaroyl-shikimate and quinate with similar efficiency, and CYP98A36 only can act on the former [24]. The activities of C3′H isoforms in wheat are plastic, and they have the ability to hydroxylate both *p*-coumaroyl ester and amides [21]. All of these demonstrated that C3′H in different species has different substrates and main functions.

Besides catalyzing the production of antioxidant substrates, C3′H also plays an important role in plant growth and development. *Arabidopsis c3′h* mutants have developmental defects with a dwarf phenotype and are vulnerable to fungal attack [25]. They also have reduced lignin content in stem, and displayed ectopic lignification in roots [26]. Repressing *C3′H* expression in potato by expressing antisense *StC3′H* under the control of the tuber-specific patatin promoter leads to a decrease in tuber yield and phenolic metabolites [27].

Because the occurrence of PPD in cassava is boosted by an oxygen burst with the over-accumulation of hydroxycoumarins, PPD symptoms can be delayed by overexpressing genes involved in ROS scavenging or interfering with the biosynthesis of hydroxycoumarin. Co-expression of the Cu/Zn superoxide dismutase (Cu/ZnSOD) and catalase (CAT) genes in cassava delayed PPD symptoms by up to 10 days [28]. The storage root-specific overexpression of the cyanide-insensitive alternative oxidase in cassava plants also delayed PPD [29]. Reduced gene expression of feruloyl CoA 6′-hydroxylase (*F6′H*), the enzyme controlling the biosynthesis of scopoletin, by RNA interference (RNAi) in cassava was shown to decrease scopoletin accumulation and PPD symptoms [30]. More recently, the ectopic expression of lysozyme HEWL driven by the vascular-specific p54/1.0 promoter was shown to attenuate the blue vascular streaking symptoms by inhibiting the accumulation of scopoletin and scopolin and delaying PPD [31]. Since the vascular discoloration symptom of PPD is strongly related to hydroxycoumarin synthesis, the goal in this study was to repress the expression of *M. esculenta C3′H* (*MeC3′H*) by the use of RNAi approach. Knockdown of *MeC3′H* significantly delayed PPD symptoms and reduced the scopoletin and scopolin contents in SRs, which validated the notion that MeC3′H played a vital role in hydroxycoumarin synthesis and demonstrated that this enzyme is a useful target for delaying PPD symptoms in cassava through genetic engineering.

## 2. Results

### 2.1. MeC3′H Shows High Homology with C3′H Members in Euphorbiaceae

The cassava *MeC3′H* gene (GeneBank Accession No.: ON605657) is 1524 bp in length and encodes a protein of 508 amino acids. To investigate the relationship between MeC3′H and other C3′Hs from dicots and monocots, a phylogenetic analysis was conducted using the protein sequences. MeC3′H was most closely related to *Hevea brasiliensis* C3′H (HbC3′H) and also showed high homology with C3′H enzymes from *Ricinus communis* (RcC3′H); all of these plants belong to the Euphorbiaceae family (Figure 1A). Similar to other C3′Hs, MeC3′H also had three conserved motifs (Figure 1B), and the third motif contained the putative substrate recognition site (SR1, indicated by a dashed line in Figure 1C) and the conserved I-helix, which is involved in oxygen binding and activation (A/G-G-X-E/D-T-T/S). The motifs between dicots and monocots also showed that C3′Hs remained relatively conservation throughout evolution.

### 2.2. The Expression of MeC3′H Is Induced during PPD

To investigate the function of MeC3′H during PPD, *MeC3′H* expression in SRs harvested from 5-month-old plants in the field was examined at different times after harvesting. After a slight decrease of *MeC3′H* expression at 24 h post-harvest, gradual upregulation was observed in the following period (Figure 2), which confirmed that *MeC3′H* transcript accumulation was induced during the PPD process.

### 2.3. MeC3′H-RNAi Transgenic Cassava Plants Develop Normally

Sequences corresponding to the third motif and flanking regions, which were found to be highly conserved based on phylogenetic and conserved motif analysis (Figure 1), were used to construct an RNAi vector. Four independent transgenic plants (C3′Hi) were produced using *Agrobacterium*-mediated transformation. After Southern blot analysis in which *Hin*dIII-digested genomic DNA was hybridized with a digoxigenin-labeled hygromycin phosphotransferase (HPT) gene probe, three lines were confirmed to have single-copy insertions, and one line was found to have multiple insertions (Figure 3A). The transcript levels of *MeC3′H* in each transgenic plant were significantly decreased to less than 50% of the level in wild-type (WT) plants (Figure 3B).

All the transgenic and WT plants grew normally in the field and developed SRs with similar weights (Figure 4), which implied that knockdown of the *MeC3′H* gene had no effect on cassava development and growth.

### 2.4. Reduced MeC3′H Expression Delays PPD Symptom Occurrence

SRs of WT and C3′Hi transgenic plants harvested at 5 months were used to evaluate PPD symptoms according to the method of the International Center for Tropical Agriculture (CIAT). SRs of WT cassava showed typical PPD symptoms with browning or blue-browning of the vascular bundles at 48 h after harvest. Symptoms became more severe at 72 h and 96 h, with dark-brown spots visible in the SRs (Figure 5A). In contrast, the SRs of C3′Hi transgenic plants had less visible brown coloration and showed weak PPD symptoms at 48 h; symptoms then gradually appeared but were less severe at 72 h and 96 h (Figure 5A). At 96 h, the level of vascular discoloration in the SRs of WT reached about 67.47%, while the levels of discoloration were only about 30.29%, 27.65%, and 27.42% in the SRs of C3′Hi transgenic plants 1, 3, and 4, respectively (Figure 5B).

Even though there was some variation in *MeC3′H* transcript levels between the three C3′Hi transgenic plants, the SRs of all these plants had lower levels of *MeC3′H* transcripts than WT SRs at all time points during the PPD process (Figure 5C). All of these results implied that MeC3′H plays a vital role in PPD by affecting the formation of black-blue deposits.

### 2.5. MeC3′H Expression Affects the Scopolin and Scopoletin Contents

To determine the effect of knockdown of *MeC3′H* expression on hydroxycoumarin accumulation in the SRs of cassava, a metabolite analysis was performed on hydroxycoumarins extracted from the SRs of cassava over a time-course of 4 days post-harvest. In SRs of WT, the contents of scopolin and scopoletin obviously increased during the PPD process (Figure 6). Scopolin increased from 20.52 μM/g FW at 0 h to a peak level of 115.74 μM/g FW at 48 h, then gradually decreased after 72 h. Scopoletin increased from 36.70 μM/g FW at 0 h to a peak level of 129.71 μM/g FW at 72 h. In contrast, the contents of scopolin at different time points in SRs of C3′Hi transgenic cassava plants were significantly lower (about 50%) than those in WT (Figure 6). The levels of scopoletin were also lower in C3′Hi transgenic plants than in WT except at 24 h and 96 h (Figure 6). The contents of esculetin and esculin were so extremely low compared with previous reports that the contents were not determined in this study, which might be caused by differences in cultivars examined and the geographical environment [6]. These results confirmed that knockdown of *MeC3′H* expression led to decreased accumulation of hydroxycoumarin metabolites in cassava SRs.

### 2.6. MeC3′H-Knockdown Cassava Shows a Reduction in Lignin Content

C3′H also is the intermediate constituent for biosynthesis of lignin and other phenylpropanoid pathways in plants, so the lignin contents were analyzed in WT and C3′Hi transgenic plants. There was no difference in the lignin contents in leaves of WT and transgenic plants, but the lignin content in the SRs of the C3′Hi-1 transgenic cassava plant (84.67 μg/mg CWM) was significantly lower than that in WT (159.72 μg/mg CWM) (Figure 6), which showed that MeC3′H also affected lignin synthesis in cassava SRs.

## 3. Discussion

PPD specifically occurs in cassava SRs, and symptoms appear within 24 h to 72 h after harvest; this deterioration decreases the shelf life of cassava SRs and reduces the quality and further application of cassava [30]. Exploring the underlying mechanism and identifying efficient and economic ways to delay PPD are of great importance. Fluorescence and secondary metabolite analyses have demonstrated that hydroxycoumarins are the main components causing vascular streaking and discolorations [6,7,8,10,11]. Cassava cultivars that accumulate hydroxycoumarins are susceptible to PPD [5], so an alternative approach for delaying PPD in cassava is to breed varieties with lower hydroxycoumarin contents. Favorable trait segregation and high heterozygosity in the cassava genome constrain the screening progress of preferential varieties with PPD tolerance [32] and traditional hybridization breeding. In recent years, there has been more success in using transgenes to decrease the contents of scopoletin and scopolin; examples include ectopic expression of HEWL to inhibit PPO activity and silencing of the expression of *F6′H* in cassava [31,32]. C3′H is considered to be a key enzyme for the synthesis of scopoletin and scopolin in plants [16,33]. In this study, we successfully generated transgenic cassava with lower scopoletin and scopolin contents in SRs by knocking down *MeC3′H* expression, which led to significantly delayed PPD in SRs.

There are three *C3′H* genes in *Arabidopsis*, namely *CYP98A3*, *CYP98A8*, and *CYP98A9*. Downregulation of *CYP98A3* (At2g40890), which plays an important role in the synthesis of scopoletin and scopolin, results in only 3% of the WT levels of scopoletin and scopolin in the roots [16,19]. Increases in scopoletin and scopolin have been observed at the early stage of PPD after harvest in cassava [6,7,11,15,31]. The induction of *MeC3′H* transcripts during PPD in WT was confirmed in this study (Figure 2), and the contents of scopoletin and scopolin both significantly increased after harvest. On the contrary, SRs of C3′Hi transgenic plants had lower scopoletin and scopolin contents during the PPD process compared with WT and showed a significant delay of PPD with less visible blue or dark discolorations, which is similar to the phenotypes of HEWL-overexpressing cassava plants [30,31]. Our study verified that MeC3′H plays an important role in PPD by regulating the synthesis of scopoletin and scopolin, and our findings also implied that scopoletin and scopolin play a substantial role in PPD symptom development [30] instead of acting as a subsidiary symptom [14,15]. It was reported that the antioxidative activities of scopoletin and scopolin might play important roles in resistance against pathogen infection and in plant defense [16,34,35,36]. Therefore, C3′Hi transgenic plants might be vulnerable to pathogens. Nevertheless, *StC3′H* antisense transgenic and control potato lines exhibited similar responses to certain bacterial pathogens [27].

In plants, hydroxycoumarin metabolites are intermediates in the phenylpropanoid pathway, and they might also play different roles in plant growth and development. In this study, C3′Hi transgenic plants exhibited a normal phenotype (Figure 4), but a decreased lignin content in SRs of C3′Hi transgenic plants was observed. In *Arabidopsis*, *c3′h* mutants exhibited a dwarf phenotype due to low lignin levels [25,37], which might be related to the collapse of many vessels due to changes of lignin biochemistry [25]. Based on these findings, it was speculated that PPD symptoms were delayed in C3′Hi transgenic plants because of a reduction of scopoletin and scopolin accumulation together with reduced lignin content leading to fewer blue or dark discolorations in the vascular tissues of cassava SRs. C3′Hi transgenic cassava had no difference in yield compared with WT under field-grown conditions, indicating that MeC3′H might be mainly responsible for the synthesis of hydroxycoumarin metabolites in cassava SRs during PPD. In potato, antisense *StC3′H* expression resulted in the reduction of tuber yield and phenolic metabolites without an obvious reduction in aboveground growth [27]. The 3′-hydroxylation step catalyzed by C3′H is regarded as the important step for the allocation of carbon in the phenylpropanoid pathway [37,38,39], so the reduction yield might be a result of metabolic perturbations affecting carbon reallocation between sink and source tissues in potato.

Variations of scopoletin and scopolin composition might be related with different functions. In this study, the contents of scopolin (5.10 μM/g, being equal to 82.08 nmol/g) and scopoletin (36.70 μM/g, being equal to 146.81 nmol/g) in SRs of cassava were significantly different from those in roots of *Arabidopsis* (1200 nmol/g of scopolin and 6 nmol/g of scopoletin) [19]. Because the roots of *Arabidopsis* are usually more easily exposed to microorganisms and physical wounding, higher levels of protective compounds such as coumarins might accumulate in roots rather than in aerial parts of the plants. In cassava, SRs mainly act as the tissue for storing carbohydrates besides absorption function such as other roots, so the contents of scopoletin and scopolin are relatively low. The scopoletin levels might also be affected by the growth environment; lower scopoletin levels have been reported for the same TMS60444 cultivar (8 ng/mg) used in our analysis [30].

*F6′H* and *C3′H* are both key genes involved in the coumarin biosynthesis pathway at different steps [35], and F6′H and C3′H catalyze hydroxylation at the 3′- and 6′-positions of the benzene ring, respectively. Downregulation of the expression of these genes in mutants strongly decreased scopoletin and scopolin levels to about 3% compared with WT [16,30,35]. During cassava PPD, *MeC3′H* was slightly induced (by 1.5-fold) in SRs of WT, and *F6′H* expression strikingly increased from barely detectable levels to approximately 10,000-fold higher levels, indicating that the inducibilities of these hydroxycoumarin synthesis pathway genes are different [30]. There is only one *MeC3′H* gene in the cassava genome, so it is inferred that it is important for maintaining normal growth and development; in contrast, four *MeF6′H* genes with divergent functions are expressed in SRs of cassava [30], which allows distinct changes of their expression. These findings indicate that it might be easier and more efficient to decrease scopoletin and scopolin levels by repressing *C3′H* expression.

## 4. Materials and Methods

### 4.1. Sequence Alignment and Phylogenetic Analysis

Protein sequences of MeC3′H and C3′Hs from other species were obtained from NCBI GenBank. The phylogenetic analysis was carried out using the neighbor-joining method in MEGA7.0 with 1000 bootstrap replications. The motifs of C3′Hs were analyzed using MEME software 5.4.1 (https://meme-suite.org/meme, accessed on 3 August 2022) [40]. DNAMAN was used to perform multiple sequence alignments.

### 4.2. Plasmid Construction and Establishment and Growth of Transgenic Cassava

The p35S::C3′H-RNAi binary vector was constructed using the pRNAi-dsAC1 plasmid backbone with AC1 replaced by a partial complementary DNA (cDNA) sequence (762 to 1119 bp) of cassava *MeC3′H* (GenBank Accession No.: ON605657). The plasmid was mobilized into *Agrobacterium tumefaciens* strain LBA4404 and then delivered into friable embryogenic callus of cassava TMS60444 to produce transgenic plants as previously reported [41].

The transgenic and WT plants were propagated in vitro in medium and then were grown in pots with nutrient soil (mixture of soil/perlite/vermiculite = 1:1:1) in a greenhouse (16 h/8 h of light/darkness, 30 °C/22 °C). For the field experiment, 2-month-old plants grown in pots were cultivated at Wushe Plantation for Transgenic Crops, Shanghai (31°13948.0099 N, 121°28912.0099 E) in May and harvested after 5 months. The phenotypes were recorded and analyzed during the growth process at regular intervals [29].

### 4.3. Molecular Verification of Transgenic Plants

Southern blot analysis was used to determine the pattern of T-DNA integration in transgenic plants according to a standard protocol with a DIG-labeled *HPT* probe [42].

Transverse slices of SRs from three 5-month-old plants at different times (0 h, 12 h, 24 h, 48 h, 72 h, and 96 h) during the PPD process and leaves harvested from 5-month-old plants were used to extract total RNA with RNA-Plant Plus Reagent (Tiangen, Beijing, China). *Me**C3′H* expressions in WT and transgenic plants were analyzed by qRT-PCR as described [43]. *β*-actin was used as the internal control. All the experiments were performed using three biological replicates. The primers were as follows:

C3′H (forward, 5′-GCAACAGAAGGCTCAAGAGG-3′; reverse, 5′-CATAGCCACC AAGCTTGACA-3′) and β-actin (forward, 5′-TGATGAGTCTGGTCCATCCA-3′; reverse, 5′-CCTCCTACGACCCAATCTCA-3′).

### 4.4. Visual PPD Evaluation

The method from the CIAT [7,10,28,44] was used to analyze the PPD process. The proximal and distal ends of SRs were cut off after harvest. The distal ends were covered by Parafilm and the proximal ends were exposed to specific conditions (22 °C to 28 °C, 70% to 80% humidity). The sections at proximal ends were harvested at 48 h, 72 h and 96 h, respectively. The levels of vascular discolorations were represented by percentages determined using ImageJ processing software 2.0 (http://rsb.info.nih.gov/ij/, accessed on 27 March 2022).

### 4.5. Determination of Scopoletin and Scopolin Contents

Scopoletin and scopolin contents were measured as previously described [6]. One gram of cassava SRs was ground with 4 mL absolute ethanol, the extracts were centrifuged for 15 min at 10,000 rpm, and the supernatants were collected. The compounds in supernatants were identified by comparing their retention time with the standard samples as controls (scopolamine, scopolidine, esculin, and esculin; Sigma, Tokyo, Japan). Determination of the compositions and quantities of compounds in the supernatant was performed with the Agilent HPLC 1200 MS Q-TOF 6520 system. HPLC separation was performed according to the following conditions: a Zorbax Extend-C18 column (3.0 mm × 50 mm, 1.8 μm), with 98% H_2_O containing 20 mmol/L acetic amonium as solvent A and 2% ACN as solvent B, at a flow rate of 0.2 mL/min. An increasing gradient (*v/v*) of solvent B was used (time (min), A:B): 0, 98:2; 2, 95:5; 5, 90:10; 15, 85:15; 18, 45:55; and 20, 0:100. The mass spectrometric parameters were as follows: mass range 40~500 m/z (MS scan rate 1.4 spectra/s), positive scan, gas (N_2_) temperature 345 °C, gas flow 9 L/min, VCap 3400 V, Fragmentor 160 V, Skimmer 64 V, Octopole RF 750 V, and Ext Dyn Standard 2 GHz (1700). 

### 4.6. Lignin Content Measurement

The lignin content was analyzed according to a modified acetyl bromide method previously described [45]. One and half milligram of cell wall material (CWM) was placed in 2 mL microtubes (one blank tube acting as a control), and 300 μL acetone was used to wash the tube wall and collect all samples at the bottom of the tube, drying acetone completely. Fresh 25% acetyl bromide (*w/w*) dissolved in glacial acetic acid (100 μL) was added into the tubes and reacted at 50 °C for 3 h. During digestion, the tubes were vortexed at 15-min intervals and cooled to room temperature. Next, 400 μL NaOH (2 M) and 70 μL hydroxylamine hydrochloride (0.5 M) were added into the solution and vortexed. The solution volume was adjusted to 2 mL with glacial acetic acid, inverted several times, and centrifuged at 10,000 rpm for 5 min. The UV spectra of the supernatants were measured against a blank prepared using the same method. The lignin content was measured by recording the absorbance at 280 nm:Lignin content (μg/mg, CWM) = (ABS/Coeff × 0.539 cm)(2 mL/weight) × 1000Coefficient (mg^−1^·mL·cm^−1^): *Arabidopsis* = 15.69.(1)

### 4.7. Statistical Analysis

All experimental data were expressed as means ± SD. The statistical analysis was carried out using Student’s *t*-test or Duncan’s multiple comparison test, using the SPSS 15.0 (for Windows) software package (SPSS Inc., Chicago, IL, USA). A value of *p* < 0.05 was considered significant.

## 5. Conclusions

In conclusion, after harvest, SRs undergo PPD with the accumulation of coumarins leading to vascular discolorations. Inhibition of *MeC3′H* expression by RNAi in cassava plants delayed the PPD process and decreased the contents of scopoletin and scopolin compared with WT. The lignin contents in C3′Hi transgenic plants decreased in parallel. Our study verified that MeC3′H plays a substantial role in PPD by affecting scolopetin and scopolin biosynthesis and also showed that MeC3′H affects lignin metabolism.

## Figures and Tables

**Figure 1 ijms-23-09231-f001:**
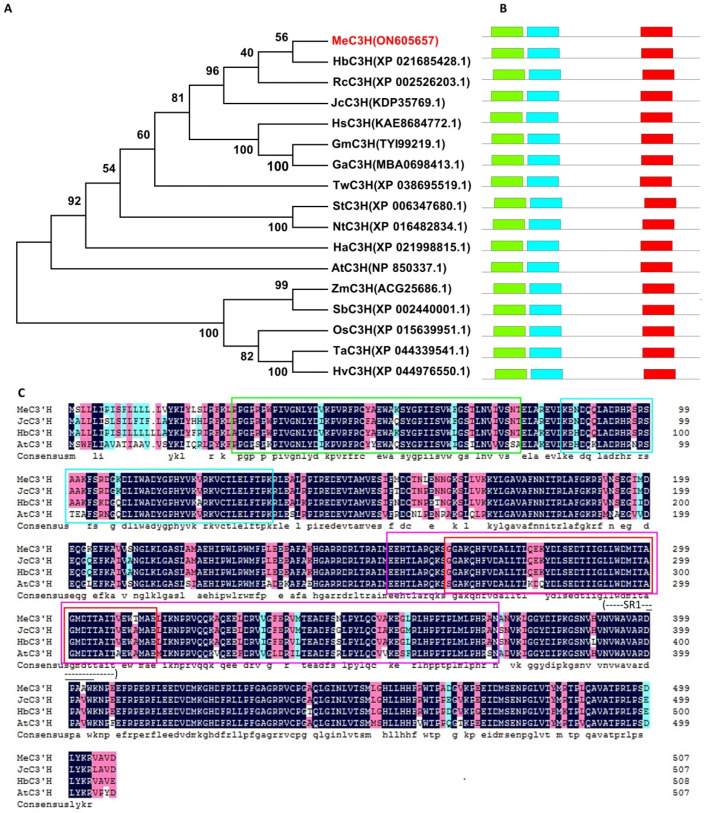
Phylogenetic tree, conserved motifs, and amino acid alignments of C3′Hs from various species. (**A**) A phylogenetic tree of C3′Hs was constructed using the Neighbor-Joining method in MEGA7.0. Bootstrap values from 1000 replicates are shown at each node. C3′Hs in the tree are from the following species: AtC3′H (*Arabidopsis thaliana*); GaC3′H (*Gossypium aridum*); GmC3′H (*Gossypium mustelinum*); HaC3′H (*Helianthus annuus*); HbC3′H (*Hevea brasiliensis*); HsC3′H (*Hibiscus syriacus*); HvC3′H (*Hordeum vulgare*); JcC3′H (*Jatropha curcas*); MeC3′H (*Manihot esculenta*); NtC3′H (*Nicotiana tabacum*); OsC3′H (*Oryza sativa*); RcC3′H (*Ricinus communis*); SbC3′H (*Sorghum bicolor*); StC3′H (*Solanum tuberosum*); TaC3′H (*Triticum aestivum*); TwC3′H (*Tripterygium wilfordii*); and ZmC3′H (*Zea mays*). (**B**) The motifs of C3′Hs were analyzed using MEME software (https://meme-suite.org/meme, accessed on 3 August 2022). The green (Motif 1), blue (Motif 2), and red (Motif 3) squares indicate Motif 1, Motif 2, and Motif 3, respectively. (**C**) Amino acid alignments of C3′H sequences from *M. esculenta*, *H. brasiliensis*, *J. curcas* and *A. thaliana*. DNAMAN was used to perform multiple sequence alignments. The green, blue, purple, and red rectangles indicate Motif 1, Motif 2, the conserved sequence used for C3′Hi construction, and Motif 3, respectively. The dashed lines indicate the putative substrate recognition site (SR1), and the continuous black line indicates the I-helix, which is involved in oxygen binding and activation (A/G-G-X-E/D-T-T/S). The black shading indicates 100% identity, and pink shading indicates 75% identity.

**Figure 2 ijms-23-09231-f002:**
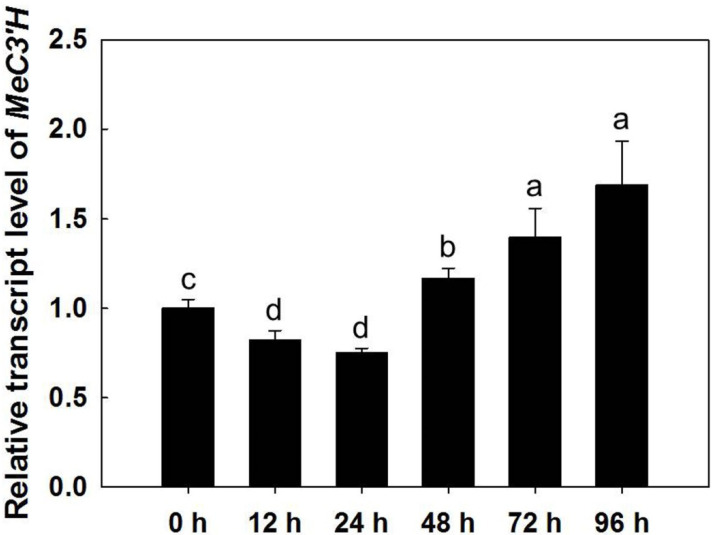
Expression of *MeC3′H* in storage roots of wild type at different times during the PPD process. Values labeled with different letters (a, b, c and d) at the different time points are significantly different according to Duncan’s multiple comparison tests at *p* < 0.05. The storage roots of plants grown for 5 months were collected to detect the expression of *MeC3′H*.

**Figure 3 ijms-23-09231-f003:**
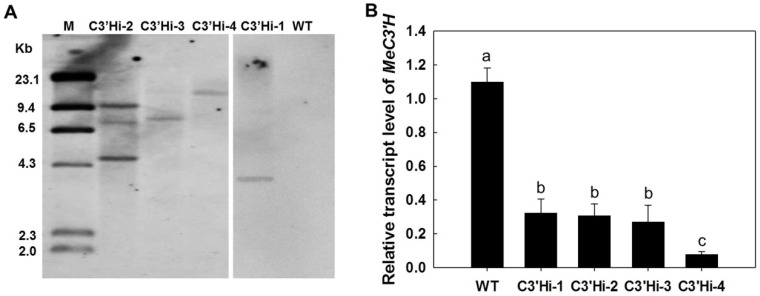
Analysis of transgenic cassava generated by *Agrobacterium*-mediated transformation. (**A**) Southern blot analysis of *Hin*dIII-digested genomic DNA from the leaves of transgenic cassava and WT using the *HPT* probe. There was only one restriction enzyme site in the T-DNA region. M: λDNA molecular weight marker; WT: wild type; C3′Hi-1–C3′Hi-4: independent transgenic lines. (**B**) Transcript levels of *MeC3′H* in transgenic plants and WT. *β*-actin was used as an internal control. Total RNA was extracted from leaves. Values labeled with different letters (a, b and c) at the different time points are significantly different according to Duncan’s multiple comparison tests at *p* < 0.05. Data are presented as means ± SD of three independent RNA samples.

**Figure 4 ijms-23-09231-f004:**
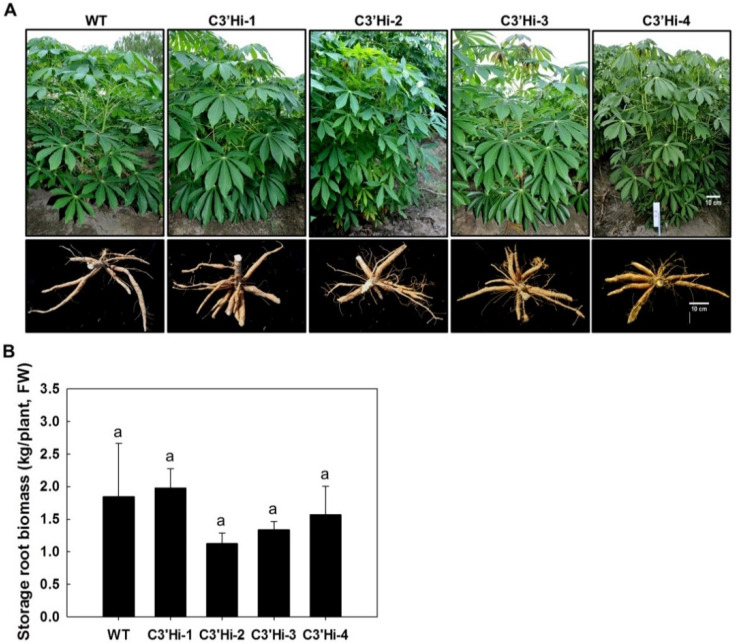
Phenotypes of C3′Hi transgenic plants and WT in the field. (**A**) Phenotypes of the aerial parts (top) and storage roots (bottom) of C3′Hi transgenic plants and WT. (**B**) Yields of storage roots from C3′Hi transgenic plants and WT. Values labeled with ‘a’ at the different time points are not significantly different according to Duncan’s multiple comparison tests at *p* < 0.05. Bars = 10 cm.

**Figure 5 ijms-23-09231-f005:**
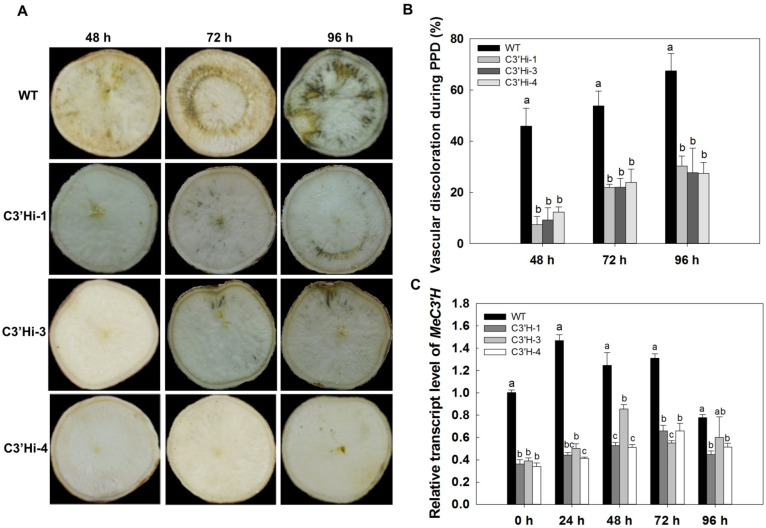
Delayed PPD occurrence in storage roots from transgenic cassava. (**A**) Visual detection of PPD occurrence using the method from the International Center for Tropical Agriculture. (**B**) The levels of vascular discoloration are shown as percentages determined using ImageJ processing software. (**C**) Lower levels of *MeC3′H* expression in transgenic cassava compared with WT was observed during PPD. Values labeled with different letters (a, b, and c) at different time points are significantly different according to Duncan’s multiple comparison tests at *p* < 0.05.

**Figure 6 ijms-23-09231-f006:**
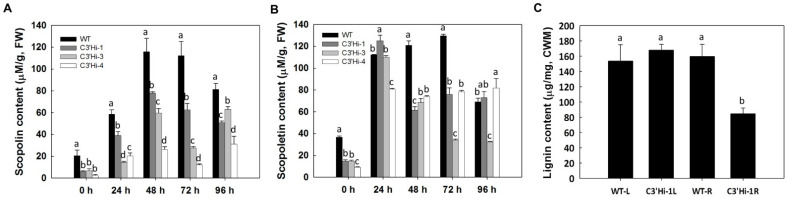
Analysis of secondary metabolite contents in SRs of transgenic plants and WT during the PPD process. (**A**) Scopoletin contents. (**B**) Scopolin contents. (**C**) The lignin content in leaves and SRs of WT and C3′Hi-1. The plants were grown in the field for 5 months, and the mature leaves and storage roots were collected for lignin analysis. FW: fresh weight; CWM: cell wall material; L: mature leaf from the fifth to seventh apical position; R: storage root. Values labeled with different letters (a, b, c and d) at the same time point are significantly different according to Duncan’s multiple comparison tests at *p* < 0.05.

## Data Availability

All the data in this study are included in this published article.

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
