# Peer review of "Knockdown of *p*-Coumaroyl Shikimate/Quinate 3′-Hydroxylase Delays the Occurrence of Post-Harvest Physiological Deterioration in Cassava Storage Roots"

_ijms, 2022, doi:10.3390/ijms23169231_

Round 1

Reviewer 1 Report

The title "Knowdown of p-coumaroyl shikimate/quinate 3'-hydroxylase delays the occurrence of post-harvest physiological deterioration in cassava storage roots" is not clear.  I assume the authors mean knockdown.

Author Response

Response to Reviewer 1 Comments

We greatly appreciate the valuable comments and suggestions from the editor and reviewer which have significantly improved the quality of our work. Below we have addressed all issues raised one by one and revised our manuscript accordingly.

Point 1: English language and style are fine/minor spell check required.

Response 1: We have asked a colleague of English native speaker to check the entire manuscript, as suggested.

Point 2: The methods description should be improved.

Response 2: We have checked and improved the experimental methods, thanks for your suggestion.

Point 3: The title "Knowdown of p-coumaroyl shikimate/quinate 3'-hydroxylase delays the occurrence of post-harvest physiological deterioration in cassava storage roots" is not clear.

Response 3: We have revised the title, thanks for your suggestion.

Reviewer 2 Report

In this paper untitled "knockdown of p-coumaroyl shikimate/quinate 3'-hydroxylase delays the occurence of post-harvest physiological deterioration in cassava storage roots", Ma et al. generated C3'H miRNA mutants and showed that while these plants exhibited normal development, they accumulated lower levels of scopolin and scopoletin and the post-harvest deterioration of the storage roots was reduced. Even if originality of the data is not high, the results are convincing and reviewer thinks the manuscript may be of interest for the readers of International Journal of Molecular Sciences. Nevertheless, reviewer thinks that authors should make some improvements before it can be published.

Line 45: authors used either CYP98A or CYP98. Is it correct?

Line 46: the end of the sentence has to be modified (rather than transferring p-coumaric...). I am not sure tha the word o-hydroxylation is right. 

Line 51: usual has to be replaced by usually

Line 55: authors probably refers to the protein and not the gene

Line 84: "in cassava" may be removed

Line 96: authors wrote tha Jatropha curcas C3'H is a close homolog of the MeC3'H. According to the tree in Fig. 1, it is very different. In addition, the bootstrap values are very low (28 and 49). I am not sur that the claimed homology is robust.

Legend of Fig. 2: I think "at the same time" must be replaced by "at the different time point"

In Fig. 3B: Why is the value for the WT different from 1?

Authors generated 4 transgenic lines (Fig. 3) but only three are analyzed in Fig. 5 and Fig.6. Is there a reason?

The expression level of C3'H in the WT (Fig. 5C) does not have the same trend than that shown in Fig. 2. Authors may comment on this discrepancy.

Line 197-199: The unit have to be modified (µM g FW).

Line 203-206: This sentence must be rephrased.

Line 209: Only line 1 was analyzed for its lignin content. What about the other lines?

What means CWM?

Line 233: the sentence has to be improved "While screening..."

Line 263: "appeared" may be replaced by "exhibited" 

Line 280: Where did the contents of scopolin and scopoletin come from? The unit is not the same as in the results. The unit line 289 is again different which render interpretation difficult. It must be improved.

Line 300: I do not understand the end of this sentence "allows comparable...". This should be rephrased.

Line 360: "EP tubes" should be replaced by "microtubes"

Titles of the axis are usually difficult to read (text too small as well as the numbers). For example, in Fig. 5 B and C.

References should be carefully checked because there are multiple errors. Some examples: Ref 3: Exp Agric. instead of Exp. Agric.; Ref12: "Inter. J. Mol. Biol." should be replaced by "Int. J. Mol. Biol."; Ref 19 and Ref23 are the same.

Author Response

Response to Reviewer 2 Comments

Point 1: In this paper untitled "knockdown of p-coumaroyl shikimate/quinate 3'-hydroxylase delays the occurence of post-harvest physiological deterioration in cassava storage roots", Ma et al. generated C3'H miRNA mutants and showed that while these plants exhibited normal development, they accumulated lower levels of scopolin and scopoletin and the post-harvest deterioration of the storage roots was reduced. Even if originality of the data is not high, the results are convincing and reviewer thinks the manuscript may be of interest for the readers of International Journal of Molecular Sciences. Nevertheless, reviewer thinks that authors should make some improvements before it can be published.

Response 1: We greatly appreciate the valuable comments and suggestions from the editor and reviewer which have significantly improved the quality of our work. Below we have addressed all issues raised one by one and revised our manuscript accordingly.

Point 2Moderate English changes required.

Response 2: We have asked a colleague of English native speaker to check the entire manuscript, as suggested.

Point 3Line 45: authors used either CYP98A or CYP98. Is it correct?

Response 3: We have confirmed that it is CYP98 here.

Point 4Line 46: the end of the sentence has to be modified (rather than transferring p-coumaric...). I am not sure that the word o-hydroxylation is right.

Response 4: We have modified the sentence in line 46, and revised o-hydroxylation into 3- hydroxylation.

Point 5:Line 51: usual has to be replaced by usually.

Response 5: We have revised it.

Point 6:Line 55: authors probably refers to the protein and not the gene.

Response 6: We have changed C3'H gene into C3'H.

Point 7:Line 84: "in cassava" may be removed.

Response 7: We have removed the words "in cassava".

Point 8:Line 96: authors wrote tha Jatropha curcas C3'H is a close homolog of the MeC3'H. According to the tree in Fig. 1, it is very different. In addition, the bootstrap values are very low (28 and 49). I am not sur that the claimed homology is robust.

Response 8: We have revised it, MeC3'H has high homology with HbC3'H and RcC3'H compared with other species.    

Point 9:Legend of Fig. 2: I think "at the same time" must be replaced by "at the different time point"

In Fig. 3B: Why is the value for the WT different from 1? Authors generated 4 transgenic lines (Fig. 3) but only three are analyzed in Fig. 5 and Fig.6. Is there a reason?

Response 9: The words "at the same time" have been replaced by "at the different time point".

In Fig. 3B, the relative expression level of WT and transgenic plants have been calculated with the formula " =POWER(2,-((B2-D2)-(C2-E2))), the values of WT are 1.141, 1.051 and 1.102, respectively. The average value is 1.098.

Even though there are four transgenic lines, Southern blot analysis showed that C3'Hi-1, C3'Hi-3 and C3'Hi-4 are single copy lines, C3'Hi-2 is the multiple copy insertion, so we have just selected three single copy insertions for further analysis.

Point 10: The expression level of C3'H in the WT (Fig. 5C) does not have the same trend than that shown in Fig. 2. Authors may comment on this discrepancy.

Response 10: We have used the different batches of samples for these two analysis. The different size, growth state and harvested time of storage roots might result into the different expression pattern. Especially for the detection in Fig. 5C, too much storage roots of WT and transgenic plants need to be collected for further analysis, we had detected the PPD of storage roots after all the samples were harvested, so the longer time had been used, which also might result into the discrepancy of expression pattern. 

Point 11: Line 197-199: The unit have to be modified (µM g FW).

Response 11: We have modified the unit according to the suggestion.

Point 12: Line 203-206: This sentence must be rephrased.

Response 12: We have rephrased the sentences.

Point 13: Line 209: Only line 1 was analyzed for its lignin content. What about the other lines?

Response 13: We are sorry for this, we haven't analyzed the lignin contents in other lines. In the following experiment, we will systematically analyze the lignin contents and the related gene expression.

Point 14: What means CWM?

Response 14: CWM means cell wall material. We have supplemented the abbreviation in line 360.

Point 15: Line 233: the sentence has to be improved "While screening..."

Response 15: We have improved the sentence.

Point 16: Line 263: "appeared" may be replaced by "exhibited" 

Response 16: We have replaced "appeared" with "exhibited".

Point 17: Line 280: Where did the contents of scopolin and scopoletin come from? The unit is not the same as in the results. The unit line 289 is again different which render interpretation difficult. It must be improved.

Response 17: The contents of scopolin and scopoletin have been detected in the storage roots of cassava. In order to compare with previous study, we have revised the description of the unit according to data.

Point 18: Line 300: I do not understand the end of this sentence "allows comparable...". This should be rephrased.

Response 18: We have changed the "comparable" to "distinct".

Point 19: Line 360: "EP tubes" should be replaced by "microtubes"

Response 19: We have replaced the "EP tubes" by "microtubes".

Point 20: Titles of the axis are usually difficult to read (text too small as well as the numbers). For example, in Fig. 5 B and C.

Response 20: We have revised the axes and the titles in Figures.

Point 21: References should be carefully checked because there are multiple errors. Some examples: Ref 3: Exp Agric. instead of Exp. Agric.; Ref12: "Inter. J. Mol. Biol." should be replaced by "Int. J. Mol. Biol."; Ref 19 and Ref23 are the same.

Response 21: We have checked all the references, thanks for your suggestions.